# Complex Organisms Must Deal with Complex Threats: How Does Amphibian Conservation Deal with Biphasic Life Cycles?

**DOI:** 10.3390/ani13101634

**Published:** 2023-05-14

**Authors:** Nadine Nolan, Matthew W. Hayward, Kaya Klop-Toker, Michael Mahony, Frank Lemckert, Alex Callen

**Affiliations:** 1Conservation Science Research Group, School of Environmental and Life Sciences, University of Newcastle, Callaghan, NSW 2308, Australia; matthew.hayward@newcastle.edu.au (M.W.H.); kaya.klop-toker@newcastle.edu.au (K.K.-T.); michael.mahony@newcastle.edu.au (M.M.); alex.callen@newcastle.edu.au (A.C.); 2Eco Logical Australia Pty Ltd., Perth, WA 6000, Australia; frank.lemckert@ecoaus.com.au

**Keywords:** amphibian, combined conservation, biphasic, eggs, larvae, metamorphs, juveniles, adults, threats, mitigation

## Abstract

**Simple Summary:**

Programs to conserve biphasic amphibians may fail if threats at different life stages are not addressed. Anthropogenic threats, such as altered hydroperiods and water pollution, exacerbate the already high natural rates of mortality at the aquatic egg and larval stage, while the terrestrial life stage is threatened by disease and habitat destruction. Threats at both of these life stages influence population viability. However, our examination of the literature suggests that studies rarely address threats to both life stages, and conservation actions rarely attempt to manage threats across the life cycle. The conservation of biphasic amphibians may thus be substantially improved by applying multiple conservation actions that deal with specific anthropogenic threats across all life stages.

**Abstract:**

The unprecedented rate of global amphibian decline is attributed to The Anthropocene, with human actions triggering the Sixth Mass Extinction Event. Amphibians have suffered some of the most extreme declines, and their lack of response to conservation actions may reflect challenges faced by taxa that exhibit biphasic life histories. There is an urgent need to ensure that conservation measures are cost-effective and yield positive outcomes. Many conservation actions have failed to meet their intended goals of bolstering populations to ensure the persistence of species into the future. We suggest that past conservation efforts have not considered how different threats influence multiple life stages of amphibians, potentially leading to suboptimal outcomes for their conservation. Our review highlights the multitude of threats amphibians face at each life stage and the conservation actions used to mitigate these threats. We also draw attention to the paucity of studies that have employed multiple actions across more than one life stage. Conservation programs for biphasic amphibians, and the research that guides them, lack a multi-pronged approach to deal with multiple threats across the lifecycle. Conservation management programs must recognise the changing threat landscape for biphasic amphibians to reduce their notoriety as the most threatened vertebrate taxa globally.

## 1. Introduction

A variety of human activities have resulted in the rapid decline in global biodiversity [1,2]. Many species are now extinct, and others are facing extinction due to the synergistic effects of these activities [3,4], including habitat destruction and degradation [5,6], changing climates [7], and the introduction of disease and invasive predators [8,9]. Scientific projections suggest that more than a third of the remaining described species will become extinct within the next 30 years if threats continue unchecked [7,10,11]. With the present rate of extinction being up to 10,000 times greater than the average natural rate [10], the multiple threats facing global biodiversity likely call for a large toolbox of conservation actions to pull species back from the brink [12,13].

Amphibians have received notoriety for being the most threatened vertebrate taxa since their first global assessment in 2004 [14]. Research suggests the most likely explanation for the current amphibian extinction crisis is disease, but declines have been exacerbated by the interaction of multiple anthropogenic threats: drought, wildfires and extreme temperature exacerbated by climate change; habitat loss, degradation and fragmentation; and the introduction of invasive species [15,16,17,18,19]. These threats reduce survival and reproductive output [20], reducing the population size [21,22] and decreasing genetic viability [23,24,25].

As many frogs (44%) exhibit a complex biphasic life cycle, they are exposed to different threats across their lifespan (Figure 1). The embryo and larval stage of biphasic amphibians are confined to aquatic environments (or very moist terrestrial habitats), with natural processes such as competition and predation causing mortality rates of up to 90% [26,27]. Anthropogenic threats within aquatic environments that may exacerbate the problem of embryo and larval survival include water pollution [28,29], changes to natural hydrology [30], pathogens [31,32], and invasive predators [33,34]. As larvae metamorphose and emerge from the water as froglets to disperse into the terrestrial environment, they are naturally prone to desiccation and predation [35,36], with the challenge of survival and successful reproduction as frogs compounded by fires [37], drought [30], habitat loss [38], and disease [39]. The combinations of these factors exert pressure on more than one life stage, thus exacerbating the potential for extinction and complicating conservation management. Hence, multiple threats to amphibian survival likely require multiple conservation actions to ensure population persistence.

Many amphibian conservation programs are not reported in the scientific literature, which is a direct contravention of the problem-solving discipline that is conservation science [40,41]. Of those programs that are reported, they largely identify poor or unexpected outcomes, reflecting the ongoing failure to arrest the current trend of amphibian decline [42,43,44]. While several authors have identified that site-based solutions do not adequately mitigate against known threats to assist amphibian recovery [45,46], a 2013 survey of more than 350 scientists and practitioners involved in amphibian conservation identified threat reduction as the most significant ingredient in successful conservation [47]. Despite substantial investment in amphibian conservation programs in the last 30 years, 41% of species belonging to the global amphibian community remain threatened by extinction [48]. This suggests that we are still a long way from an adequate representation of amphibians on the IUCN Green List of Species which benchmarks conservation success against species recovery [49], although the lack of a holistic assessment undoubtedly plays a part.

Due to the large number of threats and poor success rate of amphibian conservation, we reviewed the literature to see if employing multiple conservation treatments that focus on multiple life stages improves conservation outcomes. However, our search returned too few papers to robustly compare if using multiple actions is better than single actions. Therefore, we give an overview of how different conservation actions can target different threats to both the aquatic and terrestrial life stages, and make recommendations for how these actions can be combined to improve conservation outcomes. Lastly, we summarise the few studies that have implemented multiple conservation actions across multiple life stages and report their success.

## 2. Methods

We evaluated published scientific literature to determine the extent to which amphibian conservation studies (experimental, field observation, or review) attempted to mitigate threats at each stage of the biphasic amphibian life cycle, i.e., the aquatic stage (embryos, larvae as tadpoles) and the terrestrial stage (metamorphs, juveniles, sub-adults, and adults). A database of English-language papers published in academic journals was compiled by searching the electronic databases of scientific journals, including Web of Science, Google Scholar, and Science Direct. The keywords used for the searches included (“amphibian” OR “anuran” OR “frog”) AND (“ontogeny” OR “ontogenetic” OR “life cycle” OR “life stage”) or combinations of (“adult“ OR “juvenile” OR “sub-adult”) AND (“larvae” OR “tadpole” OR “egg”), and different combinations of (“conservation” OR “threat” OR “management” OR “mitigate” OR “eliminate” OR “eradicate” OR “preservation” OR “habitat management” OR “habitat creation” OR “habitat enhancement” OR “habitat management” OR “habitat fragmentation” OR “habitat loss” OR “adaptive management” OR “reintroduction” OR “translocation” OR “rewilding” OR “assisted colonisation” OR “disease” OR “clearing” OR “predation” OR “pollution” OR “inbreeding” OR “genetics” OR “genetic resistance” OR “captive breeding” OR “head-starting” OR “sperm cryopreservation” OR “assisted reproductive technology”). We acknowledge that keywords used in the search of the electronic databases may not reveal the extent of all studies and may be an imperfect surrogate.

The review focused on research-based conservation actions that aimed to directly bolster amphibian species and populations in the wild (in situ) and within captive populations (ex situ), and that also aimed to increase genetic diversity within populations. The categories of conservation actions considered in this review were guided by the International Union for the Conservation of Nature (IUCN, Gland, Switzerland) Red List Conservation Actions Classification Scheme (Version 2.0), and included nine broad categories: habitat protection, resource protection and management, habitat management, invasive control, habitat restoration, species management, species recovery, species re-introduction, and ex situ conservation.

## 3. Threats to the Aquatic Stage of the Amphibian Life Cycle: Embryos and Larvae

The aquatic stage of biphasic amphibians typically represents a phase of rapid growth and development [50]. For many amphibians the aquatic life cycle starts when females deposit gelatinous eggs in the water, which are externally fertilized by a male during amplexus. After a period of development, the embryos hatch and enter the larval stage [51]. The larval stage is mainly free-living and non-reproductive, and goes through metamorphosis to reach the terrestrial stage [52]. The evolutionary persistence of the biphasic life cycle and free-living larval stage suggest a strong adaptive significance to resource acquisition and growth during this phase [52]. The life histories of biphasic amphibians generally yield high numbers of progeny with very little parental care, and only a small percentage of individuals from the larval stage make it through metamorphosis [26,27,53]. As a result, human-induced pressures that further reduce the survival of the amphibian aquatic stage can have compounding catastrophic results on the viability of a population [54,55,56]. Threats include reduced hydroperiods and increased temperatures caused by climate change [54,57,58,59,60,61], predation or competition by invasive species [62,63,64], disease [39], and pollution [65,66]. Considering the natural low survival rates in the aquatic stage [26,27,53] and the emergence of compounding anthropogenic threats, the occurrence of failed recruitment has real implications for the continued persistence of wild populations [54,66].

### 3.1. Climate Change

The embryos and larvae of amphibians are confined to the aquatic habitat they are deposited into, and while many species, such as *Bufo gargarizans* [62], demonstrate evolutionary adaptation to speed-up development to escape sub-optimal environments [67,68,69], others, such as the natterjack toad (*Bufo calamita*), do not [70,71,72]. Extreme temperature changes influenced by a shifting climate can create unfavourable conditions within ponds, leading to a reduced hydroperiod and an altered pond water chemistry [73,74]. An increased frequency of drought and extended above-average temperatures can affect the survival of embryos and larval development, but this is species-dependent as a result of thermo-tolerance thresholds and certain life history characteristics specific to certain habitats [75,76,77,78]. The plasticity of the developmental period of larval amphibians in response to altered hydroperiods and water chemistry plays an important part in the long-term viability of amphibian populations [61].

A reduced hydroperiod is a key driver of recruitment failure in pond-breeding amphibians [79,80]. Studies show that the timing and length of the hydroperiod in ephemeral ponds can have an impact on the reproductive success of many amphibians [81,82,83]. When the hydroperiod in ephemeral ponds is shorter than the required development time of larvae, then reproductive success is not reached, as the larvae desiccate as the waterbody, dries [81]. As the hydroperiod of more ephemeral ponds throughout the landscape reduces, the remaining permanent ponds and their connectivity becomes increasingly important to amphibian reproduction [84,85]. However, permanent ponds are often characterised by different water chemistry, competition, predation, and resource profiles compared to ephemeral ponds [86,87,88,89].

An increased water temperature can cause mass-mortality events at the larval and embryo stages of amphibian life cycles [90]. Amphibians exposed to increased temperature profiles, consistent with current and predicted climate trends during early larval development, may have an increased rate of mortality [91], with one study demonstrating a 100% mortality rate of the common hourglass tree frog (*Polypedates cruciger*) larvae at temperatures around 34 °C, and death at metamorphosis in larvae kept at 32 °C [75]. Although some amphibian species demonstrate plasticity in their thermal tolerance, enabling them to cope with extreme temperatures, thermal tolerance and acclimation capacity vary with life stage [61]. In newly emerged larvae, acclimation capacity is low and the risk of ongoing negative effects of temperature change is high [69]. Extreme temperatures can also cause sublethal negative effects at the aquatic stage by disrupting the time to and size at metamorphosis [69,75,90,92]. The rate of development and body size at metamorphosis are vital components of amphibian fitness [90] and are species-specific. Manasee et al. [75] found that elevated temperatures delayed tadpole development time and reduced body growth in the common hourglass tree frog; however, in the Asiatic toad (*Bufo gargarizans*), warmer temperatures resulted in a shorter larval period, and a reduced body size and hind limb length [92]. Thus, the thermal landscape influences the plastic developmental traits of many ephemeral pond-breeding amphibians, and therefore shapes the growth and development [69].

Rising temperatures also negatively affect other environmental conditions, such as the decreasing dissolved oxygen levels in ponds [93], which can cause the added stress of hypoxia at the larval stage [94]. Hypoxia has been shown to cause serious abnormalities in the central nervous system of bullfrog (*Lithobates catesbeianus*) larvae, and reduce body mass and length in exposed individuals [95]. Extreme temperatures also increase bacterial blooms in ponds, giving rise to deleterious pathogens, such as heterotrophic bacteria (*Cyanobacterial lipopolysaccharide*) and cyanobacterial toxins (*microcystins*) that can affect embryo masses and cause significant liver and intestinal toxicity in larvae [90,96]. Such changes in temperature can also compromise immunity at the larvae stage, leading to increased susceptibility to infections [97,98]. The combination of these potential impacts makes climate change a significant threat to the survival of larval amphibians.

### 3.2. Invasive Species

Invasive species cause substantial environmental damage through both direct and indirect impacts on species and populations [99,100]. At the aquatic stage in the amphibian life cycle, impacts can be direct through predation or competition, as well as indirect through the modification of habitat or alteration of larval behaviour [2,62,101,102]. Invasive plants reduce the quality of amphibian aquatic habitat by altering the physical structure of aquatic vegetation. This shift in vegetation can directly and indirectly affect the aquatic stage in amphibians by disrupting food webs, changing the chemical composition of pond water, and impacting on egg deposition and clutch structure [62,102,103]. Brown et al. [62] found that the invasive plant, *Lythrum salicaria*, impacted the larvae of the American toad (*Bufo americanus*) by direct toxicity of leached tannins. Indirect negative impacts on food webs were also observed through a tadpole gut analysis, which found reduced algal communities in ponds that supported invasives compared to non-invasive plant communities [62]. Similarly, Pinero-Rodríguez et al. [103] found that the invasive floating plant (*Azolla filiculoides*) altered the chemical and physical structure of Mediterranean temporary ponds by forming a dense mat over the water surface, which decreased the pH and oxygen concentration, and increased nutrients, nitrogen, and phosphorus compounds, negatively influencing tadpole survival rates in the slow-developing western spadefoot toad (*Pelobates cultripes*).

The effects of invasive predators on amphibian populations are well documented [63,102,104]. In the presence of the predatory invasive fish, bluegill (*Lepomis macrochirus*) and largemouth bass (*Micropterus salmoides*), Rowe and Garcia [102] found a strong negative relationship with native amphibian counts, suggesting direct predation across the embryo and larval stage. Studies by Hamer [105] and Klop-Toker et al. [106] additionally found a negative relationship between the invasive mosquitofish (*Gambusia holbrooki*) and the reproduction probability of seven different frog species, including the endangered green and gold bell frog (*Litoria aurea*). Many species have defensive traits to help protect them from predators, such as increased tail fin depth or chemical recognition of predators. However, some species respond differently depending on whether they are exposed to a native predator or invasive predator that they do not have an evolutionary history with [104]. For the Iberian green frog, *Pelophylax perezi*, tadpoles detect chemical cues from native predators (dragonfly nymphs), but not invasive red swamp crayfish (*Procambarus clarkia*), demonstrating a lack of evolved predator perception to the crayfish.

Invasive anurans also negatively impact larval development rates and survivorship through exploitative competition [63,64]. Bullfrogs decreased the size of *R. boylii* metamorphs through resource competition during the larvae stage [64], and the presence of bullfrog larvae indirectly impacted the native red-legged frog (*Rana aurora*) by reducing activity levels and increasing refuge-seeking behaviour [63]. Kupferberg [64] found that predation by introduced bullfrogs (*Rana catesbeiana*) reduced the abundance of native yellow-legged frog (*Rana boylii*) larvae.

### 3.3. Diseases: Chytridiomycosis

The aquatic habitat of embryos and larvae also harbours disease-causing pathogens. Most notably, the fungal pathogen *Batrachochytrium dendrobatidis* (Bd), which causes the disease chytridiomycosis, was first linked to amphibian population declines in 1992 [107,108]. Chytridiomycosis is now considered responsible for the decline in hundreds of frog species around the world [109,110]. Bd occurs in the water and moist soil of temperate freshwater environments, and larvae become infected with its motile zoospores that penetrate the keratinised mouthparts (tooth rows and jaws) [111]. Because the area of infection is lower in larvae than the later terrestrial stages where the pathogen infects the skin [111,112], it is rarely lethal in larvae; however, it has been associated with mouthpart loss [113], and decreased activity and reduced foraging performance, causing nutrient disruptions which impact the growth and development rates [114]. Parris and Baud [115] found that exposure to Bd significantly reduced the growth and development of larvae; however, no effect was observed on survival. When larvae go through metamorphosis and start depositing keratin in other areas such as the epidermis, Bd can spread, impacting the infected metamorphosing individuals [39,116]. Metamorphosis has been identified as a highly vulnerable stage in the life cycle as immune function is reduced during this period of extreme physiological change [117,118]. As such, juveniles with immature immune systems may have a compounded susceptibility to Bd and demonstrate higher mortality rates following exposure at this stage [118,119,120].

### 3.4. Pollution and Chemical Contamination

Altered water chemistry, caused by mining, industrial, and agriculture practices, can negatively impact the aquatic stages of amphibians. Changes in pH levels and the release of coal combustion and heavy metals, such as iron, manganese, and copper, into aquatic environments have been shown to cause acute negative impacts on amphibian larvae and embryos [65,66,121,122]. Salice et al. [66] examined the population-level impacts of aquatic coal combustion residue (CCR) on the different life stages of the eastern narrow-mouth toad, (*Gastrophryne carolinensis*). Population models indicated that toads exposed to CCRs were more susceptible to decline and extinction compared to non-exposed toad populations [66]. Acidic rain and emissions of sulphur dioxide caused by industrial pollution have been found to cause impacts on the development and survival of amphibian larvae [121,123,124,125]. Farquharson et al. [121] found that prolonged exposure to decreased pH levels (increased acidity) resulted in decreased tadpole size and increased tadpole deformities. Increasing acidity was also found to delay metamorphosis in tadpoles. Mining practices and the run-off from tailing dams can release high levels of heavy metals, such as iron, into waterways [126,127], which can result in larval fatalities [122,128]. In agricultural fields, pesticides and fertilisers accumulate, and can become a source of contamination for nearby environments [129]. These chemicals are washed into rivers, lakes, and other waterways from the land [130]. Pesticides, such as glyphosate and atrazine, can cause acute and chronic effects on amphibians, including developmental effects and disruptions of the nervous system [131,132]. Due to the many potential anthropogenic sources of water pollution, this threat has the potential to impact many species that persist in non-protected areas.

## 4. Mitigating Threats at the Aquatic Stage of the Amphibian Life Cycle

### 4.1. Mitigating Climate Change

Despite the demonstrated negative effects of shortened hydroperiods and increased water temperatures on the aquatic life stage of amphibians, there are limited options to mitigate this threat, with pond creation and restoration being the most implemented action. Goldspiel et al. [85] found that by increasing the number of ephemeral and permanent ponds, amphibian populations can be bolstered, with both tadpole and frog stages benefiting. Ashpole et al. [84] attempted to bolster the abundance and occupancy of different amphibian species across multiple life stages (embryos, larvae, and adults) via constructed breeding ponds. The study by O’Brien et al. [133] aimed to conserve the great crested newt (*Triturus cristatus*) through habitat creation and restoration. The study monitored the newly created and restored ponds for the presence of embryos, larvae, and adult *T. cristatus*. It was found that 48% of new ponds were successfully colonised, with the newly created ponds now making up one quarter of *T. cristatus* breeding habitat within the study area. O’Brien et al. [133] additionally reported that one of the newly created ponds within the study area holds one of the largest breeding populations of *T. cristatus*. The aquatic stage of the amphibian life cycle is highly important, as this is the phase where most growth and development occur [50], determining fitness in individuals post-metamorphosis [69,92]. Thus, ignoring this stage of the biphasic amphibian life cycle is likely to greatly reduce the effectiveness of conservation programs focused on the terrestrial (frog) stage [55].

### 4.2. Mitigating Aquatic Invasives

The direct management of invasive predator species to protect native amphibians may eliminate strong top–down forces, where the effects of predation start at the top of the food chain and cascade down to lower trophic levels [101,102,134,135]. There are also demonstrated benefits of managing invasive plant species where the removal or reduction of bottom–up forces (where lower trophic levels affect the community structure of higher trophic levels) can provide benefits to the aquatic stage of amphibians [103]. Moreover, a reduction in the abundance of invasive aquatic plants can disrupt habitat structure, directly and indirectly improving egg and larval development and survival [102]. However, managing invasive species effectively on a large scale is extremely difficult, as successful control is generally achieved through total the eradication of individuals across a population [136].

### 4.3. Mitigating Chytrid in the Aquatic Environment

Many scientists believe that few options exist to effectively mitigate chytrid impacts [137,138]. Vaccines for chytridiomycosis have not been shown to be effective in controlling the disease in wild populations [139,140], and antifungal medication, such as itraconazole that is effective at the terrestrial stage, has not been successful for larvae [141,142]. Bosch et al. [143] used a combination of antifungal treatment in larvae with environmental chemical disinfection in five ponds across an island system, with infection eradicated in four out of the five ponds over two years. The addition of low concentrations of salt (sodium chloride) to freshwater has limited the growth and infective capacity of chytrid [144,145], and these concentrations have been shown to have no negative effects on the survival, growth, and development of larvae for at least one chytrid-susceptible amphibian species [146]. Nordheim et al. [147] also recently demonstrated that exposure to a prophylactic treatment composed of soluble chytrid metabolites in larvae can significantly lower chytrid intensity and prevalence. This may provide potential for protecting threatened amphibians in the wild [147].

### 4.4. Mitigating Pollution and Chemical Contamination

The mitigation of pollutants and chemical contamination in aquatic environments can be achieved through the use of chemical, biological, and physical remediation [148,149,150]. Techniques include ultrasonic waves, hybrid processes, bioremediation, photocatalytic degradation, adsorption, membrane separation, bio-purification systems, composite materials, ion exchange resins, carbon nanotubes, graphene, and nanocrystalline metal oxides [150,151,152,153,154]. The adsorption of pesticides onto low-cost materials can be an effective remediation technique for contaminated aquatic environments. Adsorbents based on nanoparticles and carbon are highly effective at removing pesticides from water [155]. Graphene oxide (GO) is a highly efficient adsorbent for heavy metal removal in aquatic environments [156]. This is due to its extensive oxygen functional groups, large specific area, and strong hydrophilicity [157]. However, compared with standard adsorbents, including active carbon and zeolite, GO composites are not economical [154]. The remediation of heavy metals can also be achieved by emerging nanotechnology. In comparison to conventional methods, nanoparticles or nanomaterials have been found to be very effective at removing a wide range of toxic metals from the environment [158]. These new and innovative techniques could provide promising mitigation solutions for heavy metal and pollutant contamination in the aquatic environment.

## 5. Threats at the Terrestrial Stage of the Amphibian Life Cycle: Juveniles and Adults

The terrestrial stage in the biphasic amphibian life cycle is geared towards dispersal and reproduction, and occurs after larvae undergo metamorphosis [159]. For most amphibians, this developmental change is accompanied by a radical niche shift from an aquatic to terrestrial environment, and entails changes in morphology, physiology, and behaviour [50,160]. This transformation shifts resource acquisition, and once aquatic, herbivorous larvae then turn into adult terrestrial carnivores [50]. The terrestrial stage in the amphibian life cycle faces many anthropogenic threats which impact their dispersal and reproductive ability. Among these threats are disease, habitat loss, degradation, and fragmentation, and invasive species [15,16,17,18,19].

### 5.1. Disease: Chytridiomycosis

Although infection by Bd has a limited impact at the aquatic stage of the amphibian life cycle, impacts are much more severe in juveniles and adults [39,119,137,161,162]. Recent reports suggest that Bd is responsible for the extinction of 90 amphibian species and has contributed to the decline of 501 species, making it the deadliest wildlife disease ever known [109,163]. Amphibians become infected with Bd by aquatic motile zoospores penetrating the skin of juveniles and adults [164,165]. When a zoospore enters an amphibian, it matures to form a reproductive body consisting of numerous diverging structures that spread throughout the skin. These structures impede osmoregulation, which reduces sodium levels. High infection loads can cause chytridiomycosis, leading to a cardiac arrest in susceptible species [39,164]. Newly metamorphosed amphibians are at the greatest risk of succumbing to chytrid, with studies finding juveniles exposed to the pathogen having significantly lower survival rates compared with adult frogs that were able to clear infection better [119,137,162]. Various laboratory and field studies have demonstrated the relationship between infection and environmental temperature [15,166], with elevated infection loads occurring within populations found in areas with cooler temperatures [167,168,169]. As a result of its spread across several continents and its impacts on amphibian populations, Bd is considered one of the biggest threats to amphibian populations. Other diseases can be found in amphibians, arising from infections by the *Ambystoma tigrinum* virus (ATV), Bohle iridovirus (BIV), and, *Batrachochytrium salamandrivorans* (Bsal) in salamanders [17]. While these diseases reduce survival in amphibians, the impacts they have on global populations far less than those seen by Bd in frogs.

### 5.2. Habitat Loss and Fragmentation

One of the most significant drivers of declines in amphibian biodiversity in the last century are habitat loss and fragmentation [38,170,171], as a result of agriculture, urban development, forestry, and mining [22,172]. The direct impacts on amphibian species have been widely documented, with many studies examining relationships between habitat availability, low dispersal, low reproductive success, and extinction risk [38,145,172,173]. This threat can restrict the ability of amphibians to disperse to refugia when environmental conditions are unfavourable [22,37,174]. Habitat fragmentation via changed vegetation matrices can also reduce amphibian dispersal effort to new breeding ponds, leaving suitable habitats unoccupied, thus reducing genetic diversity within a potential home range [175]. The creation of roads causes further habitat fragmentation, and vehicle collisions have been identified as a threat to the terrestrial stage of amphibians. The study by Petrovan and Schmidt [176] identifies pond-breeding amphibians, particularly the juvenile stages, as being highly vulnerable to road impacts. This can be amplified when newly metamorphosed individuals travel between aquatic habitats and terrestrial habitats and any reduction in survival at this juvenile stage can have a disproportionate impact on population dynamics [176]. The studies by Vos and Chardon [177], and Fahrig, et al. [178] confirm the negative impacts of roads on amphibian populations, finding lower abundances near high-traffic roads and a higher proportion of dead amphibians on roads with high-traffic intensity.

Habitat fragmentation at large scales can restrict amphibian dispersal during times of stress into refugia, where conditions are more favourable [174]. On a smaller scale, habitat fragmentation can destroy microrefugia, which enable amphibians to occupy and survive in climatically suitable locations during times of unfavourable conditions. Habitat fragmentation can reduce amphibian dispersal effort to new breeding ponds, even when there is no physical barrier [175,179]. Here, an amphibian’s reluctance to travel across large distances and through barriers or fragmented environments into unsuitable vegetation types to seek out better breeding ponds leaves a suitable habitat unoccupied, thus reducing genetic and species diversity within a potential home range [150]. Alternatively, when the distribution of breeding ponds is highly connected or at a low dispersal distance, amphibian dispersal reluctance and effort is reduced, ensuring that unoccupied ponds become inhabited and potentially increasing the reproductive output [150]. For amphibians, habitat connectivity plays a vital part in the local viability of populations, with some studies showing that the juvenile population is largely responsible for dispersal and therefore population connectivity [172,175,176,177,178,179,180], although adult movement is also common, but associated with breeding activity.

## 6. Mitigating Threats at the Terrestrial Stage of the Amphibian Life Cycle

### 6.1. Mitigating Chytrid within the Terrestrial Environment

Despite the overwhelming evidence of the negative impacts of chytrid, our literature review revealed that there have been few attempts to implement conservation actions across wild populations, with only a handful of studies attempting in situ actions to reduce the impacts of this disease. Three studies have manipulated different pond and water dynamics to reduce the infection load of chytrid [145,181,182]. As chytrid infection rates are positively correlated with cooler temperatures and a more neutral pH (optimal 6–7.5), Klop-Toker et al. [181] manipulated temperature and salinity levels within discrete breeding ponds to establish persistent breeding populations of the endangered green and gold bell frog *Litoria aurea*. However, the study produced mixed results, with frog populations in manipulated ponds still reporting high chytrid loads across the four years. Stockwell et al. [182] found that infection loads in frogs were negatively related to salt concentration, with species inhabiting water bodies with salinity concentrations of 3.5 ppt showing reduced infection loads compared to those exposed to less salt. The study demonstrated that exposure to sodium chloride concentrations >2 ppt significantly decreased host infection loads compared to no exposure (0 ppt).

Our literature review also revealed a handful of laboratory studies that tested actions that could reduce chytrid infections and complement in situ conservation actions. Clulow et al. [183] manipulated salinity levels in outdoor mesocosms to determine the host survival of chytrid-infected amphibians. Increased salinity reduced pathogen transmission between infected and unaffected individuals, significantly reducing mortality in salt-elevated mesocosms [183]. McMahon et al. [184] found behavioural (avoidance) and immunological (increases in lymphocyte abundance and proliferation associated with previous exposure) resistance to chytrid in three amphibian species. This study suggests that amphibians can obtain immunity to chytrid, increasing survival. Other studies have investigated factors relating to genetic resistance to chytrid in novel host species [185,186]. Kosch et al. [186] found that immunogenetic differences between captive southern corroboree frogs (*Pseudophryne corroboree*) could be used to enhance disease resistance and alleviate the threat of chytridiomycosis. Comparably, Zamudio et al. [185] found immune differences among amphibian species with different levels of resistance to chytrid infections, which was further affected by temperature and coinfection. The study highlighted the importance abiotic and biotic factors play in modulating immune response [185]. The findings of Kosch et al. [186] and Zamudio et al. [185] prove a possible long-term solution to chytridiomycosis that incorporates selectively breeding resistant propagules and translocating them back to their natural range.

### 6.2. Mitigating Habitat Fragmentation

Terrestrial habitat restoration is a promising conservation action aimed at mitigating the impacts of habitat loss and fragmentation at the terrestrial stage of amphibian species [102,187,188]. Habitat restoration focuses on returning degraded ecosystems back to their prior condition, keeping it as similar as possible to its natural state, and reversing any population or community declines [171,189]. Amphibian species experiencing a decline due to habitat loss and fragmentation benefit significantly from conservation actions that restore vegetation and corridors linking breeding and nonbreeding areas [189]. Management activities for terrestrial restoration typically target plant communities and bottom–up responses [187]. Gamble et al. [190] found that protecting and extending buffer zones around pond edges can greatly benefit juvenile and adult amphibian populations that reside in forest habitats that migrate to and/or from natal breeding ponds. To conserve amphibian populations affected by wildfire, Suriyamongkol et al. [37] investigated the use of PVC pipes as artificial refuge for the green tree frog (*Hyla cinerea*). Adults and juveniles used the pipes in both burnt and unburnt areas. This study confirmed that providing shelter sites and increasing structural complexity is a useful conservation tool that is capable of protecting and securing the survival of amphibians in severely altered post-fire habitats [37].

Aquatic habitat restoration and the creation of discrete breeding ponds is another popular and promising conservation action that bolsters amphibian populations. The construction of clustered ephemeral and permanent breeding habitats across the range of amphibians undergoing a decline has proven effective in significantly restoring amphibian populations and communities [84,181,191,192,193]. This conservation action is especially important when nearby natural ponds have been degraded and connectivity between ponds within the landscape is reduced. The large-scale restoration effort, where 38 created wetlands were examined by Lambert et al. [192], found that different amphibian species regularly occupied created water bodies with occupancy and abundance varying with species. Restored and created ponds in the study by Rothenberger et al. [191] also found a positive interaction, with strong correlations between amphibian reproductive success and pond size. The restoration of 15 ponds in the study by Hossack [193] found an increase in the number of breeding subpopulations for the Columbia spotted frogs (*Rana luteiventris*), but in another two amphibian populations, this was stationary. These studies demonstrate that the creation and restoration of breeding ponds can positively improve at least some populations and increase the occupancy and reproductive success in amphibian species.

A rapid way to increase amphibian populations after habitat restoration or creation is through translocation or the head-starting of propagules [194,195,196]. Amphibians can be translocated at any life stage from one natural area to another (wild–wild translocation), from captively bred to natural areas (captive–wild translocation), and from individuals that have been collected from the wild as embryos or tadpole and captively reared (head-starting), then released back (wild–captive–wild translocation) [195]. Translocation is considered a promising conservation action to protect amphibians from climate change and disease, especially in populations that have a very restricted home range and dispersal ability. However, the success rates associated with translocation programs for amphibians are highly variable. A review by Germano and Bishop [196] found that successful outcomes were significantly related to an increased number of released individuals, with programs releasing over 10000 individuals being mostly successful. Captive breeding programs can act as source populations for translocation or head-starting, and can help re-establish or enhance wild populations [197]. When used in combination with pond restoration or creation and invasive species management, translocation and head-starting can help rapidly bolster the amphibian abundance in aquatic habitats.

## 7. Ex Situ Conservation Actions: Managing the Frog, Not the Threat

To protect and conserve amphibian species against multiple threats, numerous ex situ actions can be implemented. Ex situ actions such as captive breeding programs, assisted reproductive technologies (ARTs), and biobanking help preserve genetic diversity and act as source populations for translocations, relocations, and reintroductions [197,198,199,200,201]. They can additionally function as insurance populations when wild populations are experiencing multiple and compounding threats, and in situ actions, such as habitat restoration and creation, need strengthening.

Captive breeding programs are generally triggered when threats to wild populations of amphibian species become acute or the overall population size becomes very small. These programs help develop populations that can function as both an insurance against future stochastic events, and as a direct supply for propagules used in head-starting or reintroduction [197]. For many critically endangered species, such as the southern corroboree frog, the only protection against extinction was a carefully employed captive breeding program, which later successfully translocated individuals back into wild populations [202]. The southern corroboree frog needed urgent conservation and protection against the imminent risk of extinction from chytrid, with the species already experiencing the effects of a small population size and an ongoing declines throughout their home ranges [203]. Lewis et al. [204] identified the value of integrating captive breeding and research as a conservation strategy for the Panamanian harlequin frogs (genus *Atelopus*) that experience drastic Bd-related declines. The captive breeding program established ex situ populations for five different threatened species within the genus *Atelopus* to mitigate disease declines through reintroduction efforts and function as insurance populations.

Assisted reproductive technologies (Figure 2), such as gamete cryopreservation, hormone therapy (the administration of reproductive hormones to induce gamete release), and in vitro fertilization (IVF), can help overcome the uncertainties surrounding captive breeding programs and the complex environmental cues associated with reproduction in amphibians [199]. For many threatened amphibian species, the combination of ARTs with captive breeding and reintroduction programs has been highly successful, including the example of the southern and northern corroboree frogs (*Pseudophryne pengilleyi*) [199,202,205]. Cryopreservation of sperm is the most effective method of storing genetic material for use in breeding and release programs, and it has the potential to significantly reduce species loss [206,207]. To date, there have been various studies into different sperm cryopreservation techniques for different threatened amphibian species around the globe, including the endangered *Atelopus zeteki* [208], *Lithobates sevosus* [209], *Litoria aurea* [201], *Andrias davidianus* [210], and *Ambystoma mexicanum* [211].

## 8. Evidence of Effect: Successful Studies That Consider Multiple Actions to Mitigate Threats at Multiple Life Stages

In environments where multiple threats are present, conservation actions that protect both the aquatic and terrestrial life stage are notprioritised. Many programs only consider single threats at a single life stage. Out of >650 amphibian conservation papers examined, we only found 20 papers that took action against threats affecting more than one life stage. We discuss in detail several studies that highlight the success multiple conservation actions can achieve when multiple life stages and threats are considered.

Pond creation and restoration, along with invasive species control, was successfully used to conserve the Great Basin spadefoot (*Spea intermontane*), the Columbia spotted frog (*Rana luteiventris*), and the Pacific tree frog (*Pseudacris regilla*) [84]. Ashpole et al. [84] increased the quantity and quality of wetland habitat by reconnecting known amphibian breeding sites with constructed and enhanced ponds. Additionally, Ashpole et al. [84] removed invasive predatory species at three sites, including the invasive fish *Carassius auratus* and the invasive American bullfrog (*Lithobates catesbeianus*). The resulting effect was colonisation by adults and early metamorphic success of the two amphibian species (*S. intermontane* and *P. regilla*) in over 60% of the enhanced and newly constructed ponds. *Spea intermontane* was considered highly successful in colonization (85% of ponds) and in production of metamorphs (observed in 61% of ponds). Comparatively, *P. regilla* colonised a similar number of ponds (71%), but was less successful in breeding, with metamorphs only observed in 33% of the ponds. Metamorphic individuals of the Columbia spotted frog were only observed twice in one pond. No detection of the blotched tiger salamander (*Ambystoma mavortium*) colonization was observed. This study demonstrates that the combined creation and restoration of breeding ponds with invasive species removal can positively improve at least some populations and increase occupancy and reproductive success in amphibian species.

The conservation program by Vignoli et al. [212] used a combination of actions to successfully reverse declines in the yellow-bellied toad (*Bombina pachypus*) in Italy. This species, commonly found in ephemeral shallow ponds, was experiencing declines throughout its distribution [212]. The main cause of the population decline was due to habitat fragmentation caused by wetland drainage for agricultural use, drought, and invasive boar wallowing in ponds. The remaining remnant population was further exposed to the threat of chytridiomycosis. Habitat restoration and creation, along with translocation of captive-bred individuals, resulted in a net increase in twenty-one released individuals, which, coupled with natural recruitment, allowed for the doubling of the original population size [212]. This is one of very few conservation efforts that used pre-action and post-action monitoring, with combined actions achieving conservation goals. Furthermore, translocated toads bred repeatedly over the years, and hence this is considered a successful translocation and habitat restoration program [212].

Cayuela et al. [213] identified that relocations, combined with captive breeding programs, can be used as promising conservation actions for some pond-breeding amphibians. The study explored ontogenetic survival differences (larvae, sub-adults, and adults) after relocation, and how this affected the long-term population viability of *Bombina variegata*. While this study identified that post-relocation survival and program success can depend on in situ threat management (i.e., predation, interspecific competition or disease), Cayuela et al. [213] did not actively control for threats across the release site. The predominant threat *B. variegata* face is habitat loss caused by agriculture and urban development [214]; however, drought and transport infrastructure are also causing population impacts [30,179]. The estimated population growth rate (λ) of the relocated *B. variegata* population was greater (2.71) than the control (1.21) and source populations (0.85), which demonstrates the success of this program in creating a self-sustaining population. Furthermore, Cayuela et al. [179] identified that adult survival is highly linked to population growth rates and relocation success; however, conservation actions that increase fecundity can significantly enhance long-term population viability. The study by Cayuela et al. [179] demonstrates that the ontogenetic stage of relocated individuals is important when assessing success, which increases when large (>1000 individual) numbers of larvae, metamorphs, subadults, and adults are released. When considering the different stages of the amphibian life cycle, an increase in both survival and fecundity can be observed, contributing to the long-term survival of a threatened species.

While the aforementioned studies have attempted to conserve amphibian species using multiple actions across different life stages, we recognise that there are other philosophical positions (such as Ockham’s Razor) that suggest that conservation resources should be prioritised towards the most feasible action, rather than focus on the complexity of a situation. However, because of the complexity and non-linearity of ecosystems, and the uniqueness assigned to the species and situation, a substantial proportion of conservation programs do not reach their intended level of success. Therefore, we argue that there is scope for conservation programs to adopt a more holistic and multi-pronged approach to species conservation by considering how they can mitigate threats and promote survival at multiple life stages.

## 9. Conclusions

The combined impacts of multiple anthropogenic threats have contributed to the biodiversity crisis of the 21st century. Biphasic organisms that have different physiologies and inhabit different environments across their life stages are arguably exposed to more threats than those that do not, and we suggest that they need different conservation actions across the life cycle to survive. In our review, we examined the extent to which the scientific literature addressed threats across life stages, and found that limited studies considered the complete life cycle of biphasic organisms. Alarmingly, only a very small number of studies applied multiple conservation actions to mitigate multiple threats, and few were able to quantitatively report performance. We thus consider that there is a real risk that programs for conserving biphasic amphibians may fail if threats at different life stages are not addressed, and that the viability of populations may be substantially improved by applying multiple conservation actions to address this.

Many conservation actions have failed to meet their intended goals of bolstering populations and ultimately ensuring the persistence of the amphibian species into the future [42,43,44]. It is possible that this is the unintended consequence of not addressing threats at both life stages. In biphasic amphibians, the embryo and larval stage are particularly vulnerable to anthropogenic threats that compromise aquatic environments, such as altered hydroperiods, water quality and competition, and predation by invasive predators. In the terrestrial environment, frogs are threatened by disease, habitat fragmentation, and shifting climates. Thus, population viability is compromised at both ends of the life cycle, whereby the progeny is compromised within waterbodies and breeding adults are compromised within terrestrial landscapes. With already naturally high rates of mortality at the larval and juvenile stage, the failure rate of amphibian conservation programs is perhaps not surprising. We provide a new perspective on the conservation of biphasic amphibians by advocating that for multiple threats, multiple action frameworks for future amphibian conservation efforts are needed. We believe the same conclusions hold true for other biphasic taxa, and addressing such challenges is essential if we are to provide future generations with the same opportunities of experiencing nature as the current generations [215].

## Figures and Tables

**Figure 1 animals-13-01634-f001:**
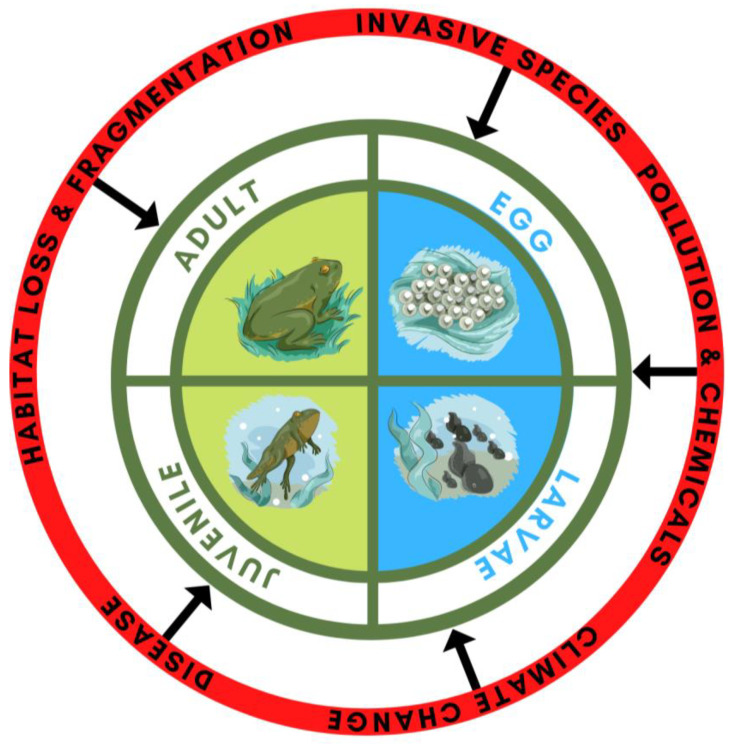
The multiple threats amphibians face throughout their life cycle. Threats include disease, invasive species, climate change, pollution and chemical contamination, habitat loss and fragmentation.

**Figure 2 animals-13-01634-f002:**
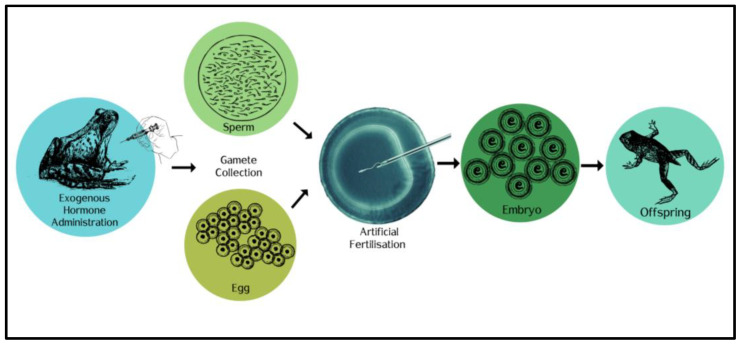
Schematic diagram of how hormone therapy and in vitro fertilization technologies can generate offspring in amphibians.

## Data Availability

Not applicable.

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
