# Peer review of "Complex Organisms Must Deal with Complex Threats: How Does Amphibian Conservation Deal with Biphasic Life Cycles?"

_animals, 2023, doi:10.3390/ani13101634_

Round 1

Reviewer 2 Report

The manuscript is a comprehensive review of conservation issues facing amphibians with a biphasic life cycle. The main purpose of the review is to explicitly describe the differences between stressors affecting aquatic stages (e.g., eggs, larvae) and those primarily affecting the terrestrial stage (e.g., juveniles, adults). Unfortunately, the authors found too few studies had considered the effects of stressors in both aquatic and terrestrial arenas to conduct an analysis, so they instead synthesize the limited available information. The manuscript is well written and concise, with appropriate breadth of coverage. I have only minor comments/clarifications, detailed below.

Line 40: add “deterioration” or “degradation” as a stressor in addition to habitat “destruction”

Figure 1: The use of the term “metamorph” makes sense to amphibian biologists, but maybe not readers in other disciplines. 'Metamorph’ is not a clearly defined life stage. Do the authors mean all individuals with terrestrial morphology that have not reached reproductive maturity (i.e., juveniles)? Or terrestrial individuals immediately subsequent to the metamorphic event, with juveniles as an omitted stage between metamorph and adult? Also, “tadpole” in my experience refers to anuran larvae to the exclusion of salamanders. Perhaps use of “larvae” would be more appropriate.

Lines 50-56: perhaps a bit of redundancy with the opening paragraph. Consider revising.

Line 58: Can the proportion of biphasic species be estimated? As a group what fraction of species demonstrate the biphasic life history strategy?

Lines 64-65: Here the definition of a metamorph is clarified, is this the definition for the figure? If so, ‘juveniles’ should be added or ‘adults’ renamed ‘terrestrial individuals’ or some similar term.

Lines 143-145: Somewhat of a strange emphasis on three species that do not dramatically alter developmental rate in response to hydroperiod variation. Why not mention the taxa that CAN change the rate, rather than cherry picking a few from the vast majority that cannot?

Line 159: Again, tadpole is a bit restrictive, use larvae?

Line 235: Is there a reason the discussion is restricted to BD? Perhaps a couple more pathogens could be mentioned, including Bsal, Ranavirus, and ATV?

Line 256: Similarly, the authors might add a few more pollutants that have been widely researched in amphibian models to their summary, e.g., atrazine, glyphosate, carbaryl, etc.

Line 279: My understanding of the literature is that adult terrestrial movements are not for ‘dispersal’ as much as ‘migration’, and that dispersal is largely carried out by emigrating terrestrial juveniles. Perhaps this can be clarified or the term “adult” could be changed to “terrestrial”

Line 342-343: This line mirrors my comment above. Please clarify the importance of distinguishing between the purposes of adult and juvenile terrestrial movements with respect to the stressors they might encounter

Line 345: Why not swap sections 4 and 5 locations? Finish the discussion of aquatic stressors prior to moving on to terrestrial

Line 536: change tense

Line 549: italicize
